# FGF21 trafficking in intact human cells revealed by cryo-electron tomography with gold nanoparticles

**Maia Azubel[1]\*, Stephen D Carter[2], Jennifer Weiszmann[3], Jun Zhang[3], Grant J Jensen[2,4], Yang Li[3,5], Roger D Kornberg[1]**

[1]Department of Structural Biology, Stanford University School of Medicine, Stanford, United States; [2]Division of Biology and Biological Engineering, California Institute of Technology, Pasadena, United States; [3]Cardiometabolic Disorders, Amgen Inc. Discovery Research, South San Francisco, United states; [4]Howard Hughes Medical Institute, California Institute of Technology, Pasadena, United states; [5]Surrozen Inc, South San Francisco, United states

**Abstract** The fibroblast growth factor FGF21 was labeled with molecularly defined gold nanoparticles (AuNPs), applied to human adipocytes, and imaged by cryo-electron tomography (cryo-ET). Most AuNPs were in pairs about 80 Å apart, on the outer cell surface. Pairs of AuNPs were also abundant inside the cells in clathrin-coated vesicles and endosomes. AuNPs were present but no longer paired in multivesicular bodies. FGF21 could thus be tracked along the endocytotic pathway. The methods developed here to visualize signaling coupled to endocytosis can be applied to a wide variety of cargo and may be extended to studies of other intracellular transactions.
DOI: https://doi.org/10.7554/eLife.43146.001

## Introduction

Imaging of cell structure has been performed using fluorescence light microscopy at modest resolution on living cells in real time, and using electron microscopy at higher resolution on fixed, embedded, sectioned material. The power of fluorescence light microscopy has been extended by super-resolution techniques (*Baddeley and Bewersdorf, 2018*), while advances in cryo-electron microscopy (cryo-EM) have yielded structures of purified proteins at near atomic resolution (*Peplow, 2017*), and have enhanced tomography of intact cells (*Oikonomou and Jensen, 2017*). Cryo-ET provides an opportunity to study proteins as they interact with a myriad of other factors (*Beck and Baumeister, 2016*; *Irobalieva et al., 2016*), often lost during protein purification. Very large multi-protein assemblies, such as ribosomes and chemoreceptor arrays, scatter electrons strongly enough that they can be recognized in electron micrographs of frozen hydrated specimens (*Briegel and Jensen, 2017*). Our approach, employing AuNP conjugates, enables the identification and image processing of most molecules and molecular assemblies, which are too small to be detected against the background of scattering from the cellular milieu. To that end, we have developed defined heavy atom clusters, targeted to individual molecules (*Azubel and Kornberg, 2016*). We report here on the application of such clusters to the fibroblast growth factor FGF21 in human primary adipocytes.

FGFs are essential in cell biology, either by their participation in cell proliferation, cell survival and cell motility (paracrine FGFs), or by their connection to metabolic processes (endocrine FGFs). These diverse activities share a common first step: binding of FGFs to cell membrane receptors. There are four genes for FGF receptors (FGFRs), which produce seven alternatively spliced variants. Paracrine and endocrine FGFs, totaling 15 and three secreted proteins, respectively, compete for binding to

**\*For correspondence:**
mazubel@stanford.edu

**eLife digest** Following a molecule's movement around a cell is a bit like looking for a needle in a haystack. Cells contain thousands of different components that can be difficult to distinguish between when viewed using a microscope. It helps to have a method to tag the molecule of interest to make it more easily visible.

Electron microscopes can capture images that reveal much finer details than traditional light microscopes. To create an electron microscope image, a high-powered beam of electrons strikes the molecules in the sample being studied. Heavier atoms scatter electrons more strongly than lighter atoms, thus, fewer electrons reach the detector and the atoms appear darker in the images. Gold atoms are heavier than the atoms that make up biological molecules (mostly carbon, nitrogen and oxygen). 'Tagging' molecules that you want to study using clusters of gold atoms would therefore help to highlight them inside cells.

Azubel et al. have now developed a method to attach gold nanoparticles to small molecules, and used the technique to track the movement of a protein called fibroblast growth factor 21 (FGF21) in human fat cells. It had previously been discovered that rats fed a high fat diet live longer and do not gain weight when treated with FGF21. Understanding how FGF21 works could therefore help researchers to develop new treatments for obesity and type II diabetes.

Azubel et al. captured many electron microscope images of cells containing tagged FGF21 proteins. This revealed that two copies of the protein work together. First, each copy of FGF21 attaches to a receptor on the surface of the cell. The two FGF21-receptor pairs bind together to form part of a larger 'complex'. The complex is engulfed by part of the nearby cell membrane, which pinches off from the rest of the membrane to form a compartment known as a vesicle. The FGF21-receptor complex stays bound together as the vesicle travels along the cell's internal skeleton. Eventually, portions of the vesicle's membrane 'bud' to form a new compartment called a multivesicular body. At this point, the FGF21 proteins and the receptors separate from each other.

Future work could build on these results in an effort to improve how we treat obesity and type II diabetes. The gold nanoparticle tracking technique developed by Azubel et al. could also be used to track other proteins using electron microscopy. This opens the way to determining the structures that proteins form when they are inside cells.

DOI: https://doi.org/10.7554/eLife.43146.002

these seven FGFRs (*Ornitz and Itoh, 2015*). Binding requires co-factors: paracrine FGFs are assisted by heparan sulfate, and endocrine FGFs by either αKlotho or βKlotho (*Kilkenny and Rocheleau, 2016*). Binding leads to FGFR dimerization and activation of FGFR tyrosine kinase activity, which triggers RAS-MAPK, PI3K-AKT, and PLCγ1 signaling cascades (*Ornitz and Itoh, 2015*). Whereas signaling is commonly thought to occur at the cell surface, it continues in endosomal locations (*Jean et al., 2010*) (*Haugsten et al., 2011*). Moreover, signaling cascades are interrupted when endocytosis is inhibited (*Yaqoob et al., 2014*). Endocytosis modulates signaling, as the specific endocytic pathway (*Mayor and Pagano, 2007*) determines whether the receptor is recycled to the cell surface or destined for degradation (*Haugsten et al., 2005*). Signaling must therefore be studied in the context of membrane internalization and vesicle trafficking.

A fundamental question regarding the activation of the signaling cascade is the stochiometry of the ternary complex (FGF-receptor-cofactor). Competing models have been proposed (*Goetz and Mohammadi, 2013*; *Yie et al., 2012*) (*Pomin, 2016*) (*Kilkenny and Rocheleau, 2016*). The crystal structure of FGF2-FGFR1c (extracellular domains D2-D3) and heparan sulfate showed a 2:2:2 ternary complex (*Schlessinger et al., 2000*). αKlotho and βKlotho differ significantly in both size and shape from heparan sulfate, and also compete with some paracrine FGFs for the same regions to bind receptors (*Goetz and Mohammadi, 2013*). Thus a different mode of binding that could lead to a different stochiometry for endocrine ternary complexes could not be ruled out. Indeed, subsequent studies of FGF21-FGFR1c-βKlotho have favored a 1:2:1 model (*Ming et al., 2012*). Most recently, the crystal structure of a 1:1:1 complex of membrane proximal portion of extracellular FGFR1c, soluble αKlotho, and FGF23 was described, and dimerization of the αKlotho complex was observed in the presence of heparan (*Chen et al., 2018*). The extracellular domain of βKlotho bound to the

C-terminus of FGF21 was also determined by X-ray crystallography, revealing a 1:1 complex, suggested to lead to an overall 2:2:2 complex (*Lee et al., 2018*).

We focus here on the FGF21-FGFR1c-βKlotho ternary complex. In recent years, FGF21 has emerged as a potential candidate for treatment of obesity and type II diabetes (*Kharitonenkov and DiMarchi, 2015*). Pleiotropy of FGF21 includes effects on glucose and lipid metabolism in adipocyte tissue (*Degirolamo et al., 2016*). FGF21 signals through FGFR1c, FGFR2c and FGFR3c, provided that βKlotho is accessible (*Kilkenny and Rocheleau, 2016*). Both FGFR1c and βKlotho are endogenously expressed in adipocyte tissue.

The pathway of FGF21-FGFR1c-βKlotho complex internalization remains an open question. Evidence for both clathrin-dependent (Jean et al.) and clathrin-independent (*Haugsten et al., 2011*) pathways, for different combinations of FGF and FGFR, has been presented. Regarding the FGF21-FGFR1c-βKlotho complex, dynamin-dependent endocytosis has been suggested (*Yaqoob et al., 2014*). However, dynamin has been found associated to both clathrin-dependent and clathrin-independent endocytosis (*Mayor and Pagano, 2007*).

With the use of gold-labeled FGF21 (AuNP-FGF21) and cryo-ET, we captured different states of activation, internalization, and traffic of the FGF21-FGFR1c-βKlotho ternary complex, from binding and complex formation at the cell surface, to coated pits, to coated vesicles, to endosomes, and finally, to multivesicular bodies, in which the complexes were disrupted. These observations are clearly indicative of clathrin-dependent endocytosis. Finally, subtomogram averaging and helical reconstruction revealed structures of other important components, including putative AAA + ATPases, actin filaments, and microtubules, giving a three-dimensional picture of the entire pathway.

## Results

### FGF21-FGFR1c-βKlotho ternary complex in membrane vesicles

A 144-gold atom nanoparticle (AuNP) was conjugated with an FGF21 variant bearing a surface-exposed cysteine residue (*Xu et al., 2013*), as described (*Azubel and Kornberg, 2016*). Interaction in ternary complexes was assessed using membrane preparations from three cell sources: parental CHO cells, in which neither FGFR1c nor βKlotho are expressed; transformed CHO cells overexpressing βKlotho and FGFR1c or only βKlotho; and human primary adipocytes, in which βKlotho and FGFR1c are endogenously expressed. Vesicles were treated at 4°C with either AuNP-FGF21 or a gold-labeled single chain antibody fragment (AuNP-scFv) that binds βKlotho, and washed to remove unbound gold conjugate. Grids for cryo-EM were prepared by plunge-freezing. Micrographs of vesicles from parental CHO cells membrane preparations treated with AuNP-FGF21 showed no associated AuNPs, whereas micrographs of vesicles from primary adipocytes membrane preparations treated with AuNP-FGF21 showed pairs of AuNPs (*Figure 1—figure supplement 1*). AuNPs were distinguishable from other particles because of an effect of the contrast transfer function, producing a bright halo around the strongly scattering gold core (*Figure 1—figure supplement 2*). Pairing of particles cannot be determined from 2D images alone, as two particles in close proximity in the x-y plane may be far apart in z. Tilt series were therefore collected for membrane preparations from CHO cells overexpressing βKlotho and FGFR1c treated with AuNP-FGF21, followed by tomographic reconstruction, showing that 85% of AuNPs were in true pairs (*Figure 1—figure supplement 3a–g*), indicative of two copies of FGF21 in the receptor complex.

Treatment of these CHO membrane vesicles with AuNP-scFv against βKlotho also resulted in a high percentage of pairs of particles (*Figure 1—figure supplement 3h*), indicative of an overall 2:2:2 stoichometry for the receptor complex. When membrane preparations from CHO cells expressing only βKlotho were treated with AuNP-scFv against βKlotho, pairs of particles were not observed (*Figure 1—figure supplement 3j*), showing that βKlotho did not dimerize on its own. When the same vesicles were treated with AuNP-FGF21, however, pairs of particles were again observed (*Figure 1—figure supplement 3i*). Either two molecules of FGF21 bind to one βKlotho, or FGF21 induces dimerization of overexpressed βKlotho, even in the absence of receptor.

Because overexpression of FGFR1c and βKlotho may lead to receptor auto-activation (*Sørensen et al., 2006*), and with a view to studies on intact cells (see below), we repeated the analysis with AuNP-FGF21 on membrane preparations from human adipocytes. As before, tilt series

were collected, followed by tomographic reconstruction, revealing 89% of AuNPs in true pairs, with an average separation (center-to-center distance) of 80 ± 15 Å (*Figure 1*). With use of the AuNPs to improve the alignment of the tilt series (*Figure 1—figure supplement 4*), protein densities on both inner and outer surfaces of the membrane were revealed (*Figure 1a* and *Figure 1—figure supplement 4c*).

## FGF21-FGFR1c-βKlotho complex on the surface of intact cells

A key requirement for extension of the analysis to intact cells is sufficient thinness of the cells for cryo-EM. CHO cells were not well suited in this regard, but cytoplasmic regions of adipocyte cells grown on Holey-Carbon Au mesh grids were as thin as 200–300 nm near the cell periphery (*Figure 2* and *Figure 2—figure supplement 1*). As in the case of vesicles from CHO and adipocyte cells membrane preparations, most AuNP particles were in pairs (88%) on the adipocyte cell surface (*Figure 3*). AuNP pairs showed a tendency to cluster, consistent with previous reports of clustering of FGF receptors from immunofluorescence studies with anti-FGFR antibodies (*Gao et al., 2015*). AuNP pairs were found in areas surrounding filipodia and, most notably, above invaginations of the cell surface membrane with clathrin nets beneath (*Figures 2* and *3*). The occurrence of most AuNP-

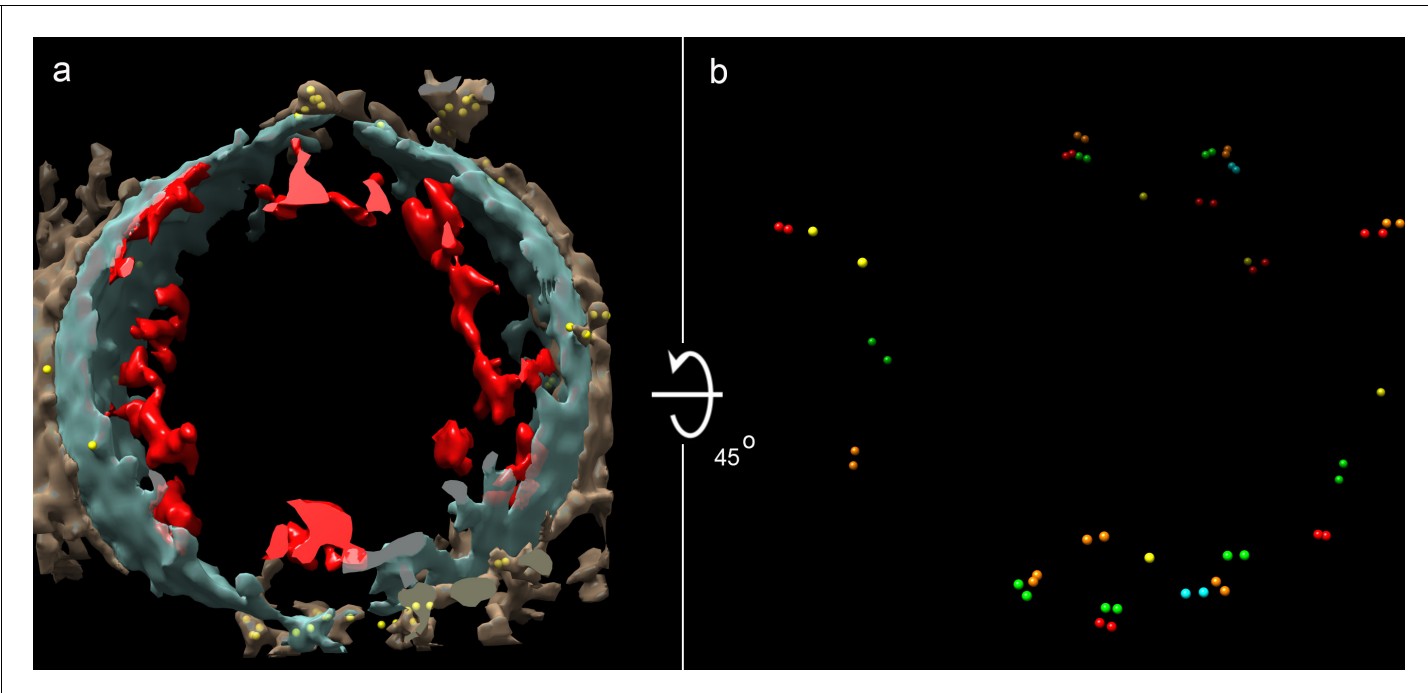

**Figure 1.** Cryo-ET of a vesicle from human adipocytes membrane preparations, treated with AuNP-FGF21 conjugate. (a) Isosurface rendering of tomographic reconstruction, with membrane density in blue-gray, density on the outer surface of the membrane in brown, density on the inner surface of the membrane in red, and AuNPs in yellow. (b) Same as (a), with all membrane and membrane-associated density removed, with different colors to distinguish pairs of AuNPs, and with rotation of 45° from the view in (a) for better visualization of AuNPs.
DOI: https://doi.org/10.7554/eLife.43146.003

The following figure supplements are available for figure 1:

**Figure supplement 1.** Cryo-EM images of membrane preparations treated with AuNP-FGF21 conjugate.
DOI: https://doi.org/10.7554/eLife.43146.004

**Figure supplement 2.** AuNPs characteristic footprint in a tomogram slice.
DOI: https://doi.org/10.7554/eLife.43146.005

**Figure supplement 3.** Predominance of AuNP pairs on the surface of vesicles from membrane preparations form CHO cells overexpressing FGFR1c and βKlotho.
DOI: https://doi.org/10.7554/eLife.43146.006

**Figure supplement 4.** Comparison of tomographic reconstruction using different markers as fiducials.
DOI: https://doi.org/10.7554/eLife.43146.007

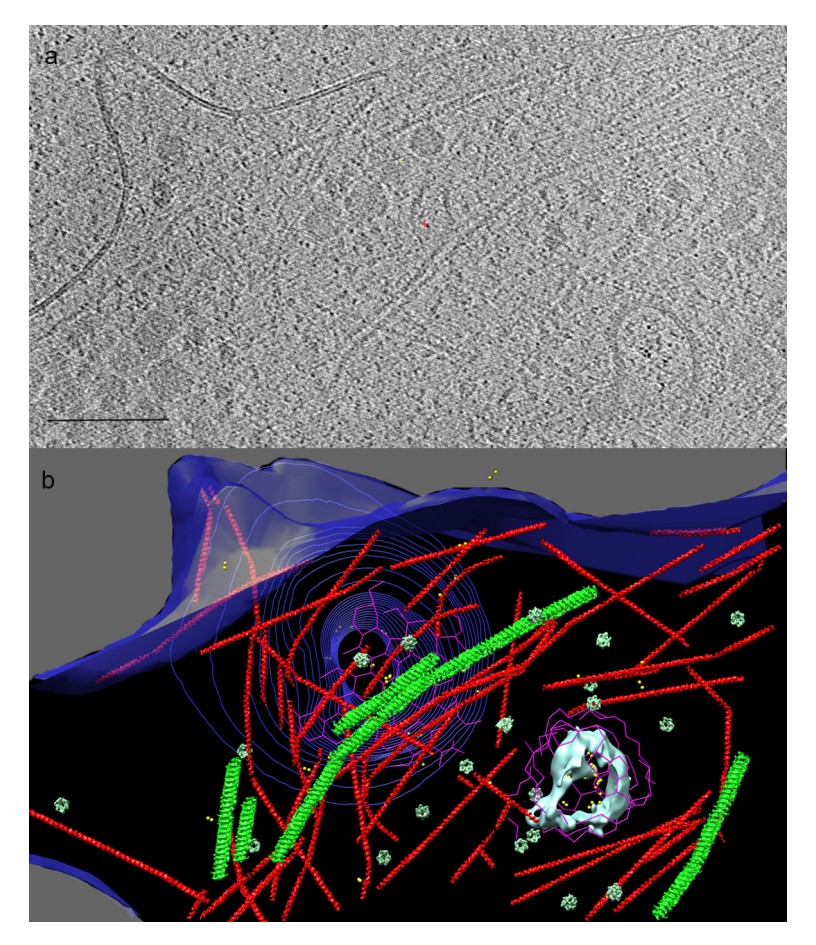

**Figure 2.** Cryo-ET of human adipocyte cell treated with AuNP-FGF21 conjugate. (a) Slice of tomogram showing a region near the cell periphery. Bar 100 nm. (b) 3D tomographic data, with the plasma membrane in blue (invagination of the membrane, viewed from inside the cell, represented by contours), isosurface rendering of a coated vesicle membrane in cyan, clathrin in magenta, actin in red and microtubules in green (substituted with helical reconstructions from *Figure 4*), hexameric rings (putative p97 AAA+ ATPAse) in emerald (substituted with subtomogram averages from *Figure 4*), and AuNPs in yellow.

DOI: https://doi.org/10.7554/eLife.43146.008

The following figure supplements are available for figure 2:

**Figure supplement 1.** Viability of human adipocyte cells transferred to EM grids, and thinness at the periphery after plunge-freezing.

DOI: https://doi.org/10.7554/eLife.43146.009

**Figure supplement 2.** Polymerization of actin filament in a y-shape provides force for membrane deformation (*Kaksonen et al., 2006*).

DOI: https://doi.org/10.7554/eLife.43146.010

FGF21 in pairs pertains to the stoichiometry of the ternary complex. Our findings are suggestive of the occurrence of 2:2:2 FGF21-FGFR1c-βKlotho complexes in vivo.

## Cytoplasmic structures and the FGF21 endocytotic pathway

A number of familiar structures were visible in the tomograms of adipocyte cells (*Figures 2* and *3*): membranes (both cell surface and vesicular), clathrin nets, actin filaments, microtubules, and hexameric rings. The resolution of the tomograms was sufficient to distinguish intercalating legs of neighboring clathrin triskelions (*Fotin et al., 2004*) (*Figure 4a*). Actin filaments and microtubules were confirmed by helical reconstruction and docking high-resolution structures into the

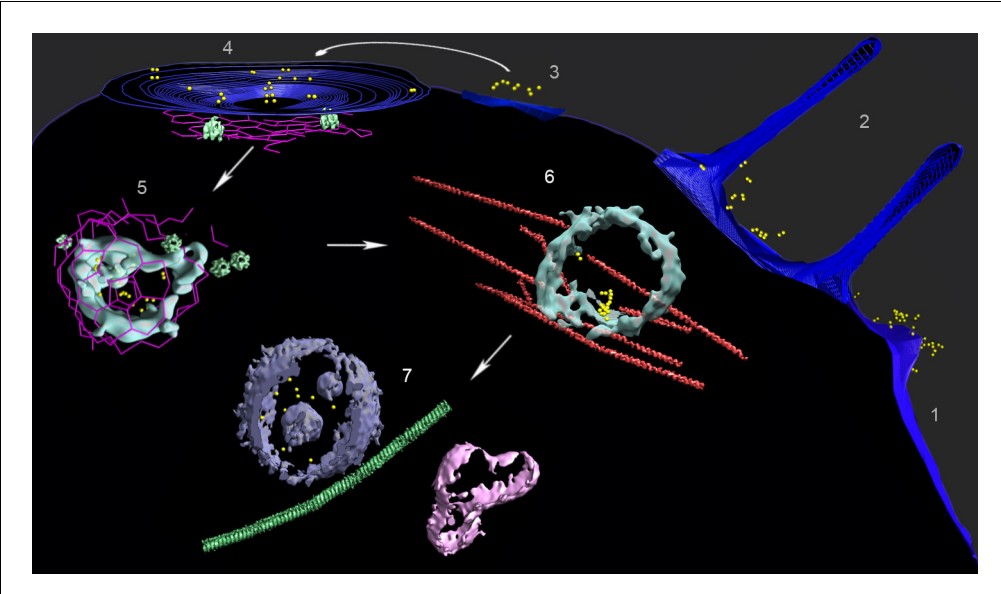

**Figure 3.** Multiple locations of FGF21-FGFR1c-βKlotho ternary complex in human adipocyte cells. A composite image from several tomograms, with the cell surface membrane in blue, isosurface renderings of coated vesicle and endosomal membranes in cyan, isosurface renderings of a multivesicular body (MVB) and other vesicle membranes in violet and pink, clathrin in magenta, actin and microtubules in red and green (substituted with helical reconstructions from *Figure 4*), hexameric rings (substituted with subtomogram averages from *Figure 4*; putative p97 AAA+ ATPAse) in emerald, and AuNPs in yellow. Tomograms collected following treatment with AuNP-FGF21 for 1 h at 4°C show (1) a lamellopodium decorated with clusters of AuNP pairs, (2) filopodia surrounding clusters of AuNP pairs, (3) clusters of AuNP pairs on the cell surface, (4) AuNP pairs clustered in a coated pit, and (5) a clathrin-coated vesicle. Hexameric rings (putative p97 AAA+ ATPAse) are abundant in the vicinity of clathrin. A tomogram following treatment with AuNP-FGF21 for 1 h at 37°C shows an endosome associated with actin filaments (6) and a tomogram following treatment with AuNP-FGF21 overnight at 37°C shows a microtubule between an MVB and another vesicle (7). The arrows indicate a possible order of events, not an actual sequence; regions numbered 1–7 were taken from different tomograms.

DOI: https://doi.org/10.7554/eLife.43146.011

The following figure supplement is available for figure 3:

**Figure supplement 1.** Activation and internalization cycle.

DOI: https://doi.org/10.7554/eLife.43146.012

reconstructions (*Figure 4b*). Hexameric rings, averaged from subtomograms, corresponded in outline and dimensions to the p97 AAA+ ATPase, although NSF and Vps4p, with similar structures, could not be excluded (*Figure 4c*).

When grids were exposed to the AuNP-FGF21 conjugate for 1 h at 4°C and transferred to 22°C before freezing, AuNP pairs were observed in clathrin-coated vesicles, about 100 nm in diameter (*Figure 2*, *Figure 3* and *Figure 3—figure supplement 1*), similar in size to clathrin-coated vesicles isolated from cells, but larger and less regular in shape than vesicles assembled in vitro (*Kirchhausen et al., 2014*). After 1 h at 37°C, AuNP pairs were observed in endosomes (*Figure 3* and *Figure 3—figure supplement 1d*). Not only were almost all AuNPs paired in both clathrin-coated vesicles and endosomes (89% and 88%, respectively), but they were also invariably adjacent to the inner membrane surface, pointing to persistence of the FGF21-FGFR1c-βKlotho complex. Finally, after overnight incubation at 37°C, AuNPs were observed in multivesicular bodies (MVBs). Among 44 AuNPs observed inside five MVBs in different cells, no two AuNPs were closer than 250 Å to one another. AuNPs in MVBs were not only unpaired but also unassociated with the vesicle membranes, indicating the disruption of the FGF21-FGFR1c-βKlotho complex in MVBs (*Figure 3* and *Figure 3—figure supplement 1f*).

Our findings demonstrate a clathrin-dependent pathway, and point to accessory factors in the process. Thus, clathrin pits were seen to be associated with abundant actin filaments, including

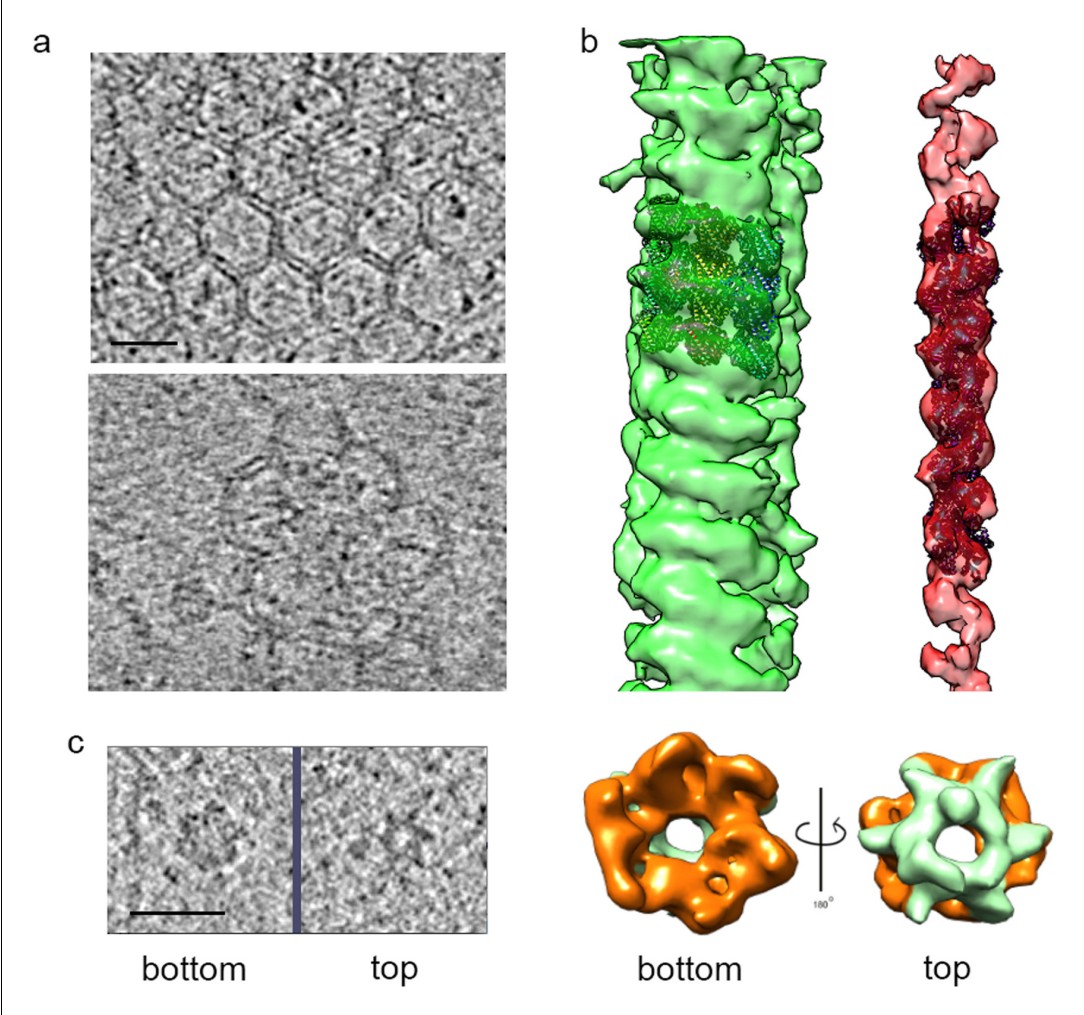

**Figure 4.** Structures identified in tomograms of human adipocyte cells. (**a**) Tomogram slices showing a clathrin net (top panel) and cage (lower panel). (**b**) Helical reconstructions of densities attributed to microtubules (green) and actin filaments (red) with high-resolution structures (PDB:IDs 3JAK and 3B5U, respectively) manually docked in the densities. (**c**) Left panel, bottom and top tomogram slices of an individual hexameric ring particle; right panel, hexameric ring subtomogram average rotated by 180°, showing, bottom (orange), and top (emerald) sides. Bars 20 nm.
DOI: https://doi.org/10.7554/eLife.43146.013

y-shaped filaments (*Figure 2* and *Figure 2—figure supplement 2*), and with hexameric rings (*Figures 2* and *3*). Clathrin nets were clearly resolved in 11 tomograms coming from nine different cells. In all cases the nets were surrounded by y-shaped actin. In 10 of the 11 tomograms, at least one hexameric ring was found within 50 nm of the net, and hexameric rings were observed in all cases if the search was expanded to 75 nm from the net. The number of hexameric rings within 75 nm varied among nets from one to 21. The association of hexameric rings with clathrin nets was supported by the orientation of the rings. The bottom surface of the rings was larger than the top surface (*Figure 4c*) and the bottom was always oriented toward clathrin (*Figure 5*). Our findings are in keeping with the literature regarding the role of actin filaments and of y-shaped filaments in clathrin-mediated endocytosis (*Kaksonen et al., 2006*), and also in keeping with the literature regarding p97-clathrin interaction and the involvement of p97 in endosomal sorting (*Meyer et al., 2012*). Our findings go further, showing persistence of the FGF21-FGFR1c- βKlotho complex in endosomes, and disruption of the complex in MVBs. The example of an MVB shown here lies in proximity to a long microtubule (*Figure 3* and *Figure 3—figure supplement 1f*). As some FGFs travel all the way to the nucleus (*Sørensen et al., 2006*), and membrane vesicles are transported along microtubules, the MVB may be involved in transport of FGF to the nucleus.

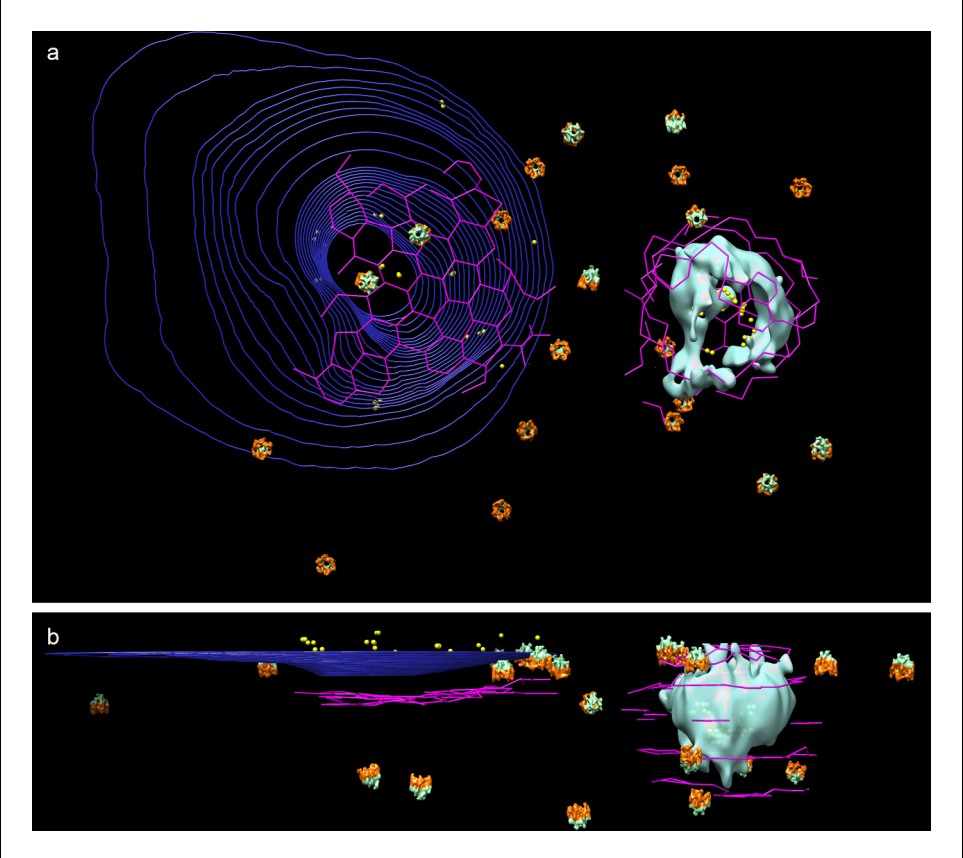

**Figure 5.** Orientation and proximity of hexameric rings to clathrin nets. (**a**) Top view and (**b**) side view of 3D tomogramographic data from cell shown in *Figure 2*, with dual color hexameric ring as in *Figure 4*. (Microtubules and actin filament have been removed for clarity).

DOI: https://doi.org/10.7554/eLife.43146.014

## Discussion

Our results from imaging AuNPs in human adipocytes by cryo-ET are of both mechanistic and methodological significance. They contribute to the emerging picture of the FGF signalling mechanism and trafficking inside cells. They show that two copies of FGF21 are present in the FGF21-FGFR1c-β Klotho ternary complex in cells, and that two copies of βKlotho are present as well, pointing to an overall 2:2:2 stochiometry. Second, FGF21-FGFR1c-βKlotho complexes undergo clathrin-dependent endocytosis. Information from multiple tomograms shows that the ternary complexes undergo clathrin-dependent endocytosis and gives a three-dimensional picture of the entire pathway. The same approach can be applied to other FGFs. Further study of both endocrine and paracrine FGFs would shed light on the complex regulation of FGFRs-induced signaling cascades.

With regard to methodological significance, our findings extend previous investigations by EM tomography of plastic-embedded sections and by cryo-EM of protein-receptor complexes in liposomes, performed with the use of commercial gold nanoparticle preparations (*He et al., 2008*; *He et al., 2009*). In the future, AuNPs of different sizes (*Azubel et al., 2014*; *Azubel et al., 2017*) conjugated with different antibodies may be used to track multiple components of a receptor complex at the same time. The approach may be used not only for tracking a variety of cargos but also, by the introduction of AuNP-scFv conjugates in cells, for studies of other intracellular transactions.

## Materials and methods

### Bioconjugation

E38C-FGF21 (*Xu et al., 2013*) and a single chain antibody fragment (scFv) against βKlotho were conjugated with 3MBA-Au144 nanoparticles (NPs) (*Azubel et al., 2017*) as described (*Azubel and Kornberg, 2016*) with minor modifications. Briefly, 200 µM E38C-FGF21 or 34 µM anti-βKlotho scFv were reduced with 1 mM TCEP for 1 h at 37°C. Reduced E38C-FGF21 was incubated on ice for 15 min, and reduced anti-βKlotho scFv was incubated for 45 min at 37°C, in the presence of twofold excess of 3MBA-Au144 NPs in both cases. Conjugates were passivated by treatment with 2.5 mM glutathione (GSH) for 30 min on ice (AuE38C-FGF21) or 45 min at 37°C (anti-βKlotho scFv). Passivated conjugates were run in a 10% glycerol, 12% polyacrylamide gel in Tris-borate-EDTA buffer at 150 V. The gel band corresponding to the conjugate was excised, and crushed and soaked overnight in PBS.

### Cell membrane preparation

AM-1/D Chinese Hamster Ovary (CHO) cells stably expressing both human βKlotho and human FGFR1c (Amgen proprietary cell line derived from CHO cells previously characterized (*Hecht et al., 2012*; *Shi et al., 2018*)) were suspended in 50 ml buffer containing 10 mM HEPES pH 7.5, 100 mM NaCl, 1 mM EDTA, and one tablet protease inhibitor (Roche). Cells were lysed by Dounce Homogenization (30 strokes on ice), followed by a spin at 1000 rpm for 10 min. Supernatant was transferred to a 50 ml centrifuge tube and volume was brought up to 40 ml before centrifugation at 16,000 rpm for 30 min. The pellet was resuspended in 1 ml buffer (10 mM HEPES pH 7.5, 100 mM NaCl, 1 mM EDTA). 10 µg of anti-βKlotho were added followed by incubation at room temperature for 2–3 h. 100 µl 50% slurry protein A beads were added and sample was rotated for 1 h at room temperature. Beads were let to settle down and washed with 10 mM HEPES pH 7.5, 100 mM NaCl, 1 mM EDTA twice. 10 µl Caspase three were added and the sample was incubated overnight at 4°C . 1 ml buffer (10 mM HEPES pH 7.5, 100 mM NaCl, 1 mM EDTA) was added and the sample was transferred to a centrifugation tube for a 30 min spin at 16,000 rpm. The pellet was washed twice, resuspended in 40 µl buffer (10 mM HEPES pH 7.5, 100 mM NaCl, 1 mM EDTA) and stored at −80°C.

7-day differentiated human adipocyte cells were suspended in 50 ml of PBS buffer containing one tablet protease inhibitor (Roche). Cells were lysed by Dounce Homogenization (30 strokes on ice), followed by centrifugation at 1000 rpm for 10 min. Supernatant was transferred to a 50 ml centrifuge tube and volume was brought up to 40 ml before centrifugation at 16,000 rpm for 30 min. The pellet was then resuspended in 40 µl PBS and stored at −80°C.

### Labeling of membrane preparations and vitrification

Membrane preparations (~5 mg/ml) from 12 different experiments were incubated with either AuE38C-FGF21 (0.03 mg/ml) or anti-βKlotho scFv (0.03 mg/ml) on ice for 30 min. The sample was centrifugated and washed with 1X PBS three times, or until the supernatant was colorless. 2.5 µl resuspended membranes were mixed with 0.5 µl 10 nm BSA Gold Tracer (EMS, Haltfield, PA, USA) before applying to glow discharged 200 mesh copper R2/2 Quantifoil grids (Quantifoil Micro Tools GmbH, Jena, Germany). Blotting and plunge-freezing into liquid ethane (at −178°C) were performed with a Leica EM GP (Leica Microsystems, Wetzlar, Germany) set to 5 s pre-blotting time, 6 s blotting time, no post-blotting time, 22°C and 90% humidity.

### Cell growth, labeling and vitrification

One vial of Cryoperserved Human Subcutaneous Preadipocyte cells (Zen Bio, NC, USA) was thawed by immersing in a 37°C water bath and gently shaking. Cells were transferred to a 50 ml tube containing 9 ml of pre-warmed Subcutaneous Preadipocyte Growth Medium (PM-1) (Zen Bio, NC, USA). Cells were centrifuged for 3 min at 1200 rpm. Medium was aspirated, and cells were resuspended in 5 ml PM-1 and transferred to a 75 cm$^2$ flask containing 10 ml of pre-warmed PM-1. Cells were grown in an incubator at 37°C in the presence of 5% CO$_2$, for 24 h, or until they were confluent. PM-1 was aspirated and 15 ml of Adipocyte Differentiation Medium (DM-2) (Zen Bio, NC, USA) was added. Differentiation proceeded for 5–7 d in an incubator at 37°C in the presence of 5% CO$_2$. Medium was aspirated and cells were washed with 10 ml pre-warmed 1X PBS, before adding 3 ml pre-warmed CellStripper (Corning, VA, USA). The flask was put back into a 37°C incubator for 5–10

min, or until the cells lifted off the plate. Cells were washed off with 7 ml of 1X PBS, collected in a 50 ml tube, and centrifugated for 3 min at 1200 rpm. Cells were resuspended in DM-2 at a density of ~$10^5$ cells/ml and plated in six-well plates, containing three to four pre-treated 10 nm BSA Gold Tracer (EMS, Haltfield, PA, USA) and fibronectin-coated 200 mesh gold R2/2 London finder Quantifoil grids (Quantifoil Micro Tools GmbH, Jena, Germany) per well. After overnight incubation at 37°C in the presence of 5% $CO_2$, the grids were placed upside down in a nine-well Teflon plate containing 30 µl drops of 35 µM AuE38C-FGF21, incubated on ice, or at room temperature, or 37°C for 1 h, or at 37°C overnight, and washed with 1X PBS. Grids were mounted onto Leica EM GP (Leica Microsystems, Wetzlar, Germany) so grids could be blotted from the reverse side. Before blotting and plunge-freezing, 3 µl of 10 nm BSA Gold Tracer (EMS, Haltfield, PA, USA) were added. Blotting and plunge-freezing into liquid ethane (at −180°C) were performed with a Leica EM GP (Leica Microsystems, Wetzlar, Germany) set to 2 s pre-blotting time, 4 s blotting time, no post-blotting time, 22°C and 95% humidity. Cells grown on grids from more than 20 experiments were taken for cryo-ET data collection.

## Cryo-ET data collection

Tilt series were collected either on a FEI (Eindhoven, The Netherlands) Tecnai F20 FEG transmission electron microscope operating at 200 kV, or on a FEI (Eindhoven, The Netherlands) F30 G2 Polara FEG transmission electron microscope operating at 300 kV and equipped with an energy filter (slit width 20 eV for higher magnifications; Gatan, Inc.). Images were recorded using a 4k × 4k K2 Summit direct detector (Gatan, Inc.) operating in the electron counting mode. Tilt series were recorded using SerialEM (*Mastronarde, 2005*) software at magnifications with corresponding pixel sizes ranging from 1.28 to 2.42 Å. Either a bidirectional or a dose-symmetric tilt schemes (*Hagen et al., 2017*) were implemented from −60° to +60° with an increment of 2° at 2–6 µm underfocus, and total dose around 120 e⁻/Å².

## Cryo-ET data processing

Tilt-series were aligned and processed with the IMOD software package (*Kremer et al., 1996*). After binning the aligned tilt series by threefold, reconstructions into 3D tomograms were done with back projection, which helps to unequivocally identify Au nanoparticles, and with SIRT (Simultaneous Iterative Reconstruction Technique) for increased contrast.

Subtomogram 3D-averaging and helical reconstruction were performed using PEET software package (*Heumann et al., 2011*). Initial segmentation was done with IMOD software package (*Kremer et al., 1996*) and Chimera software package (*Pettersen et al., 2004*) was used for visualization and docking of pdb structures into density maps.

## Acknowledgements

This research was supported by NIH grants AI 21144 to RDK and R35 122588 to GJJ. We thank Amgen Department of Protein Sciences for providing some of the reagents used in this study. We thank Dr. E P Geiduschek for discussion and comments on the manuscript.

## Additional information

### Competing interests

Jennifer Weiszmann, Jun Zhang, Yang Li: Employee of Amgen at the time the study was conducted. There are no other competing financial interests to declare. The other authors declare that no competing interests exist.

### Funding

| Funder | Grant reference number | Author |
| --- | --- | --- |
| National Institutes of Health | AI 21144 | Roger D Kornberg |
| National Institutes of Health | R35 122588 | Grant J Jensen |

The funders had role in study design, and the decision to submit the work for publication.

### Author contributions

Maia Azubel, Conceptualization, Data curation, Formal analysis, Supervision, Validation, Investigation, Visualization, Methodology, Writing—original draft, Project administration, Writing—review and editing; Stephen D Carter, Investigation, Methodology, Writing—review and editing; Jennifer Weiszmann, Resources, Investigation, Visualization, Methodology, Writing—review and editing; Jun Zhang, Resources, Writing—review and editing; Grant J Jensen, Conceptualization, Resources, Funding acquisition, Writing—review and editing; Yang Li, Conceptualization, Resources, Supervision, Funding acquisition, Methodology, Writing—review and editing; Roger D Kornberg, Conceptualization, Resources, Supervision, Writing—original draft, Writing—review and editing

### Author ORCIDs

Maia Azubel (iD) http://orcid.org/0000-0002-1584-2695
Stephen D Carter (iD) http://orcid.org/0000-0002-4237-4276
Grant J Jensen (iD) https://orcid.org/0000-0003-1556-4864

### Decision letter and Author response

Decision letter https://doi.org/10.7554/eLife.43146.017
Author response https://doi.org/10.7554/eLife.43146.018

## Additional files

### Supplementary files

• Transparent reporting form
DOI: https://doi.org/10.7554/eLife.43146.015

### Data availability

All data is provided in the manuscript and supporting files.

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
