## [Decision Letter]

Thank you for submitting your article "Tracking FGF21 traffic in intact human cells by cryo-electron tomography" for consideration by *eLife*. Your article has been reviewed by three peer reviewers, one of whom is a member of our Board of Reviewing Editors, and the evaluation has been overseen by Vivek Malhotra as the Senior Editor. The reviewers have opted to remain anonymous.

The reviewers have discussed the reviews with one another and the Reviewing Editor has drafted this decision to help you prepare a revised submission.

Summary:

The reviewers all agree that this study represents a significant technical achievement, which should be published upon completion of the revisions requested. However, there was also a general consensus that publication in the 'Tools and Resources' category, as opposed to the 'Research Article' category, seems more appropriate, due to the limited conceptual advance.

Essential revisions:

The work from Azubel et al. provides one of the most detailed examples of how a growth factor receptor engages an extracellular ligand and is subsequently transported from the cell surface via clathrin-mediated endocytosis to arrive at a multivesicular body, where the ligand dissociates. From these studies, it is now clear that FGF21 binds to its receptor and co-receptor with 2:2:2 stoichiometry (in cultured cells), and the complex remains assembled while trafficking via vesicular intermediates along cytoskeletal elements until reaching a mature, multivesicular compartment. The approaches chosen (cryo-EM combined with the use of gold nanoparticles) are ideal for studying this process. However, there is a lack of quantification throughout the study – in its present form, it is not even feasible to know how many times experiments were repeated to draw the conclusions made. Prior to publication, this issue must be addressed.

Specifically, the electron microscopy would be much more compelling if the authors supported their observations with quantitative analysis throughout the manuscript. They begin to do this in Figure 1—figure supplement 3 with the percentage of paired nanoparticles but subsequent statements are not backed by similar rigor. For example, what percent of nanoparticles were paired in coated vesicles and endosomes (subsection “Cytoplasmic structures and the FGF21 endocytotic pathway”) compared to MVBs, how often were clathrin pits associated with Y-shaped filaments and hexameric rings, and how many occurrences of hexameric rings oriented toward clathrin were observed? Even though pairs of Klotho were not observed without ligand or receptor (subsection “FGF21-FGFR1c-βKlotho ternary complex in membrane vesicles”, second paragraph) how many particles were observed and how many samples were analyzed? Also, what attempts were made to quantify the incidence of FGF21 gold nanoparticle uptake in CHO cells?

The addition of quantitative measurements for each figure should be included in the revised manuscript, and the manuscript should then be resubmitted under the 'Tools and Resources' category.

---

## [Author Response]

Essential revisions:The work from Azubel et al. provides one of the most detailed examples of how a growth factor receptor engages an extracellular ligand and is subsequently transported from the cell surface via clathrin-mediated endocytosis to arrive at a multivesicular body, where the ligand dissociates. From these studies, it is now clear that FGF21 binds to its receptor and co-receptor with 2:2:2 stoichiometry (in cultured cells), and the complex remains assembled while trafficking via vesicular intermediates along cytoskeletal elements until reaching a mature, multivesicular compartment. The approaches chosen (cryo-EM combined with the use of gold nanoparticles) are ideal for studying this process. However, there is a lack of quantification throughout the study – in its present form, it is not even feasible to know how many times experiments were repeated to draw the conclusions made. Prior to publication, this issue must be addressed.

We have incorporated quantification analysis in the revised manuscript.

The number of experiments performed for labeling of membrane preparations (12 experiments), which were subsequently used for vitrification and cryo-ET data collection, is cited in the Materials and methods (subsection “Cell growth, labeling and vitrification”). The number of tomograms, from where AuNPs were counted, collected for each case (4-5 tomograms) is indicated in the figure legend (Figure 3—figure supplement 1 legend).

For intact cells, the number of repetitions of the treatment of adipocytes with AuNP-FGF21 exceeded 20 experiments, and it is now noted (subsection “Cell growth, labeling and vitrification”).

Specifically, the electron microscopy would be much more compelling if the authors supported their observations with quantitative analysis throughout the manuscript. They begin to do this in Figure 1—figure supplement 3 with the percentage of paired nanoparticles but subsequent statements are not backed by similar rigor. For example, what percent of nanoparticles were paired in coated vesicles and endosomes (subsection “Cytoplasmic structures and the FGF21 endocytotic pathway”) compared to MVBs.

Percentages of AuNPs paired in clathrin-coated vesicles and endosomes are 89% and 88%, respectively (subsection “Cytoplasmic structures and the FGF21 endocytotic pathway”, second paragraph). No pairs of AuNPs were observed inside MVBs.

How often were clathrin pits associated with Y-shaped filaments?

Every clathrin net was surrounded by Y-shaped actin (11 tomograms coming from 9 cells – two cells had two patches of clathrin separated by more than 15µm). The tomograms were reconstructed from tilt series collected in 5 different sessions (subsection “Cytoplasmic structures and the FGF21 endocytotic pathway”, last paragraph.

How often were clathrin pits associated with hexameric rings?

10 out of 11 had at least 1 ring within 50 nm from the net. 11 out 11 had at least 1 ring within 75 nm from the net (subsection “Cytoplasmic structures and the FGF21 endocytotic pathway”, last paragraph).

And how many occurrences of hexameric rings oriented toward clathrin were observed?

The number of rings close to each net varied from 1-21. All rings were oriented (subsection “Cytoplasmic structures and the FGF21 endocytotic pathway”, last paragraph).

Even though pairs of Klotho were not observed without ligand or receptor (subsection “FGF21-FGFR1c-βKlotho ternary complex in membrane vesicles”, second paragraph) how many particles were observed and how many samples were analyzed?

103 AuNPs were counted from 5 different tomograms (figure legend Figure 1—figure supplement 3).

Also, what attempts were made to quantify the incidence of FGF21 gold nanoparticle uptake in CHO cells?

We previously pointed to the advantage of adipocytes over CHO cell. Whereas primary adipocytes represent a minimal perturbed system, artifacts created due to overexpression in CHO cells might lead to autodimerization of the receptor (subsection “FGF21-FGFR1c-βKlotho ternary complex in membrane vesicles”, second paragraph). Furthermore, in the revised manuscript we explain that CHO cells are not thin enough for cryo-ET (subsection “FGF21-FGFR1c-βKlotho complex on the surface of intact cells”).